# Advances in the Early Diagnosis of Pancreatic Ductal Adenocarcinoma and Premalignant Pancreatic Lesions

**DOI:** 10.3390/biomedicines11061687

**Published:** 2023-06-11

**Authors:** Reiko Yamada, Junya Tsuboi, Yumi Murashima, Takamitsu Tanaka, Kenji Nose, Hayato Nakagawa

**Affiliations:** Department of Gastroenterology and Hepatology, School of Medicine, Mie University, Tsu 514-8507, Japan; t-junya0128@med.mie-u.ac.jp (J.T.); ymurashi@med.mie-u.ac.jp (Y.M.); tanakatakamitsu@med.mie-u.ac.jp (T.T.); kenji-nose@med.mie-u.ac.jp (K.N.); nakagawah@med.mie-u.ac.jp (H.N.)

**Keywords:** pancreatic cancer, diagnostics, biomarkers, hereditary conditions, EUS, EUS-FNA, EUS-FNB

## Abstract

Pancreatic cancer is one of the most lethal human malignancies, in part because it is often diagnosed at late stages when surgery and systemic therapies are either unfeasible or ineffective. Therefore, diagnosing pancreatic cancer in earlier stages is important for effective treatment. However, because the signs and symptoms may be nonspecific and not apparent until the disease is at a late stage, the timely diagnoses of pancreatic cancer can be difficult to achieve. Recent studies have shown that selective screening and increased usage of biomarkers could improve the early diagnosis of pancreatic cancer. In this review, we discuss recent advancements in the early detection of pancreatic ductal carcinoma and precancerous lesions. These include innovations in imaging modalities, the diagnostic utility of various biomarkers, biopsy techniques, and population-based surveillance approaches. Additionally, we discuss how machine learning methods are being applied to develop integrated methods of identifying individuals at high risk of developing pancreatic disease. In the future, the overall survival of pancreatic cancer patients could be improved by the development and adoption of these new methods and techniques.

## 1. Introduction

Pancreatic cancer is a relatively rare cancer. However, patients with pancreatic cancer are generally diagnosed with late-stage or metastatic disease, which limits their eligibility for surgery; moreover, systemic treatments do not dramatically extend overall survival in patients with late-stage pancreatic cancer [1,2]. Thus, while pancreatic cancer is only responsible for approximately 3% of all diagnosed malignancies, it accounts for approximately 10% of all cancer-related deaths. When resectable, the primary surgical treatments for pancreatic ductal adenocarcinoma (PDAC) are pancreaticoduodenectomy and distal pancreatectomy, but these techniques are associated with relatively high recurrence rates [3]. Given the difficulty of early detection and the lack of effective treatment methods, the 5-year survival rate for pancreatic cancer has remained below 10% for the past two decades [4,5,6,7,8].

There are two main forms of pancreatic cancer, which are grouped in accordance with the cell types of origin. Exocrine pancreatic cancers account for approximately 95% of all pancreatic lesions and develop from the cells that make and secrete enzymes that facilitate the breakdown of carbohydrates, fats, and proteins in the gastrointestinal tract. Among the exocrine pancreatic cancers, approximately 90% are adenocarcinomas, which are generally thought to arise from the epithelial (ductal) cells that line the glands through which the secretory enzymes pass, although there is evidence that other exocrine cell types can transdifferentiate into duct cells that ultimately develop into adenocarcinomas [9,10]. Such cases are referred to as PDACs and account for approximately 90% of all pancreatic malignancies. Rarely (up to 4% of exocrine cases), pancreatic cancer presents as adenosquamous carcinoma, which is more aggressive and has a worse prognosis. These tumors show the characteristics of both ductal adenocarcinoma and squamous cell carcinoma. Acinar cell carcinoma is a form of pancreatic cancer that arises from the cells that produce the secretory enzymes and accounts for approximately 2% of exocrine cases. The second major class of pancreatic cancer is pancreatic neuroendocrine tumors (also termed islet cell tumors), which arise from cells that make up the endocrine gland of the pancreas, which secretes insulin, gastrin, and glucagon into the serum to regulate glucose uptake and metabolism. Neuroendocrine cancers are rare, accounting for <5% of pancreatic malignancies. Furthermore, neuroendocrine cancers sometimes cause hormonal symptoms (sometimes at onset and sometimes later in the course of the disease). Thus, treatment for hormonal symptoms may be necessary in addition to treatment for cancer. Additionally, there are functional differences in neuroendocrine tumors based on which hormone (insulin, gastrin, or glucagon) is overproduced. Many pancreatic cancers arise from precancerous lesions, and acinar-to-ductal metaplasia (ADM) is thought to be the primary cause of these premalignant pancreatic lesions [11]. ADM occurs in response to pancreatic injury or acute pancreatitis and, while reversible, becomes irreversible if chronic pancreatitis or a *KRAS* mutation is present [12].

The various types of pancreatic cancer have generally similar diagnostic criteria and are typically diagnosed through a combination of imaging tests (such as computed tomography (CT) or magnetic resonance imaging (MRI)), endoscopic ultrasound (EUS), blood tests, and biopsies. However, these diagnostic tests are typically performed late in the course of the disease because of a relative lack of early symptoms. While, in theory, screening the overall population for pancreatic cancer could help identify cases earlier, this approach is not feasible because of the high cost and low specificity of the diagnostic tests that are currently available. Another approach would be to screen only those individuals at high risk of developing this malignancy, although attempts to identify risk factors associated with pancreatic cancer have met with limited success. In this review, we discuss the current approaches to diagnosing pancreatic cancer and identifying high-risk individuals, and we highlight promising new research that could provide the ability to diagnose pancreatic cancer at stages where better treatment options and patient outcomes are possible.

## 2. Current Approaches to Diagnosing and Predicting Pancreatic Cancer

### 2.1. Current Approaches to Diagnosing Pancreatic Cancer

Pancreatic cancer can be particularly difficult to diagnose because patients typically have nonspecific symptoms. The symptoms that patients do experience—such as pain in the abdomen or spine—are often indicators of late-stage disease. Patients are often diagnosed with pancreatic cancer when they experience symptoms resulting from metastases, and approximately 50% of cases exhibit metastatic disease at the time of diagnosis [13]. The clinical features that can lead to suspicion of pancreatic cancer include new-onset diabetes, jaundice, dark urine, the aforementioned pain in the abdomen or back, and any combination of nausea/weight loss/poor appetite. Unfortunately, as many of these manifestations are common to several clinical conditions, multiple visits to the doctor’s office might be required to achieve a diagnosis. Ultimately, most pancreatic tumors are diagnosed through imaging and cytology/biopsy; however, some alternative diagnostic methods have been reported.

The primary imaging modalities currently used to diagnose pancreatic cancer are CT, MRI, and EUS. Among these methods, CT is the most widely available and best-validated tool for imaging PDAC. Several national and international cancer guidelines suggest using CT as an initial imaging modality for cases of suspected pancreatic cancer. Moreover, a dedicated CT protocol was developed specifically for pancreatic lesions [14]. This protocol provides a standardized template for reporting that specifies the terminology to be used for solid pancreatic neoplasms, with the aim of improving the patient-management decision-making process and optimizing treatment recommendations. A meta-analysis of different imaging modalities found that the sensitivity and specificity of CT for detecting pancreatic lesions were both approximately 90% [15]. CT can also detect local atrophy/fatty metamorphosis of the pancreatic parenchyma at the early stages of disease [16,17]. MRI allows the detection of the morphological changes of the pancreatic duct and pancreas parenchyma, which can facilitate the detection of pancreatic lesions at earlier stages, although the previously mentioned meta-analysis found it to be equally sensitive and specific as CT (sensitivity and specificity of approximately 90%) [15]. Thus, because of its increased cost and limited availability, MRI is not as widely used as CT for diagnosing pancreatic cancer. Finally, EUS allows detailed, high-resolution imaging of the pancreas and surrounding vessels and lymph nodes. Ikemoto et al. reported that EUS enabled a significantly higher rate of detection in patients with stage IA tumors than CT or MRI and that EUS detected masses in more than half of the patients in whom a mass was not detected by CT or MRI [18]. It was also reported that EUS has a diagnostic sensitivity of 94.4% for detecting small PDACs (<20 mm) [19]. Yasuda et al. reported that, among 132 patients with risk factors for PDAC without masses detected on CT, pancreatic tumors were detected by EUS in 3 patients [20]. Clinical guidelines issued by the Japan Pancreas Society in 2019 stated that EUS should be performed at institutions with highly skilled operators [21]. Most importantly, EUS can be paired with fine-needle aspiration (FNA) or fine-needle biopsy (FNB) to facilitate a cytopathological diagnosis. The accuracy of both EUS-FNA and EUS-FNB for diagnosing pancreatic cancer is extremely high, with a recent analysis finding that of both to be >80% [22,23]. EUS-FNA/FNB can also be useful in cases where masses are indistinct on other imaging platforms. The imaging features of PDAC at different stages are summarized in Figure 1.

In addition to diagnosing pancreatic cancer via the cytology/pathology of EUS-FNA/FNB specimens, the cytological analysis of the pancreatic juice obtained via ERCP can be used to help diagnose pancreatic malignancies [24]. The background cytological or pathological features of pancreatic neoplasms can also be helpful for distinguishing different types of pancreatic lesions. For example, desmoplastic stroma and the presence of cancer-associated fibroblasts are background features of PDAC; conversely, a mucinous background is associated with intraductal papillary neoplasms.

### 2.2. Current Approaches for Identifying Individuals at High Risk of Developing Pancreatic Cancer

While the modalities mentioned above are highly sensitive and specific for identifying lesions within the pancreas, a long-standing problem in the field is applying these imaging modalities to patients before their disease has become unresectable or metastatic. Although there are guidelines for identifying individuals who are at high risk of developing pancreatic cancer [25] based on risk factors including genetics/family history [26,27] and environmental/lifestyle factors [28], there are limitations to their utility [1,29]. The difficulties in identifying high-risk individuals are influenced by the biology of the disease, wherein >80% of pancreatic cancers have nonhereditary *KRAS* mutations; thus, most cases are the result of somatic mutations [30]. However, between 10% and 15% of pancreatic cancers are associated with known inherited mutations and/or show familial trends. Pancreatic cancer patients typically develop symptoms during the later stages of disease progression; thus, early detection programs have been developed for asymptomatic individuals deemed to be at high risk of developing pancreatic cancer. High-risk factors include, but are not limited to, familial pancreatic cancer, inherited syndromes such as Peutz–Jeghers syndrome and Lynch disease, familial atypical multiple mole melanoma syndrome, hereditary pancreatitis, and *PALB2* and *BRCA2* mutations (Table 1). The detection programs suggest different timings for the initial imaging screenings depending on individual risk factors, with the frequency of future screening depending on the initial findings.

Although markers in pancreatic juice can potentially be used to diagnose pancreatic cancer, serum markers have shown limited utility for the early diagnosis of lesions. The serum markers carbohydrate antigen 19-9 (CA19-9) and carcinoembryonic antigen (CEA) have some efficacy for identifying high-risk individuals [33]. CA19-9 is the only serum biomarker for PDAC approved by the U.S. Food and Drug Administration, but it yields false-negative and false-positive results in certain cases [34]. Additionally, CA19-9 levels can increase under other conditions, such as in the presence of benign tumors, inflammatory masses, diabetes, or acute pancreatitis [35,36,37]. Nevertheless, recent preclinical studies showed that CA19-9 may be mechanistically linked with the development of PDAC [38], so it remains a notable marker for the minimally invasive detection of pancreatic cancer.

## 3. Improvements to Current Diagnostic Approaches

The past decade has seen several promising developments in the early diagnosis of pancreatic cancer; while most are in preclinical or early phase testing, in the coming years, these new methods are likely to be incorporated into clinical practice. These advancements include modifications or improvements to current radiography and imaging modalities as well as the development of novel biomarkers and analytical techniques for biopsy samples.

### 3.1. Innovations in Interpreting Radiomics and Imaging Findings

While CT and cytology are both currently used to diagnose pancreatic cancer, the findings need to be interpreted by experienced radiologists and pathologists, respectively, and preferably by those with specific expertise in pancreatic cancer. Unfortunately, such tools and expertise are not common to all hospital settings. To make the best diagnostic approaches universally available, artificial intelligence is being applied to identify lesions and determine tumor stage/grade. Briefly, these methods involve the use of machine learning (in which a computer learns to complete a task or make a decision based on a source dataset) or deep learning (which uses an artificial neural network to learn how to make better decisions based on the same type of dataset) algorithms to identify multimodal features of PDAC to improve diagnostics.

The majority of such studies have developed algorithms for CT imaging data. In 2022, Qureshi et al. used a naïve Bayes classifier to extract features from abdominal CT scans and developed an algorithm to identify individuals at high risk of developing PDAC [39]. This study involved retrospectively analyzing CT imaging data from healthy control, pre-diagnostic, and diagnostic groups, then analyzing features that could predict high-risk cases. The model developed in this study had an 86% average classification accuracy. Another machine learning algorithm that was developed from CT data was shown to detect pancreatic lesions prior to clinical diagnoses [40]. Similar approaches have also been applied to develop tools that can predict the tumor grade from CT imaging. For example, in a study that included 91 PDAC patients grouped by tumor grade, Tikhonova et al. used one-factor logistic models and LASSO regression to develop an algorithm for predicting PDAC grade from radiomics features of CT imaging [41]. Specifically, the CT images were assessed for the presence of 62 “texture features”, which were then evaluated for their ability to discriminate between cancerous and healthy tissue. The algorithm showed a good ability to discriminate grades 2 and 3 PDAC using texture features from CT; however, the authors cautioned that different scanning protocols can influence the results of the algorithm. Similarly, another study used machine learning to develop and validate a 12-factor radiomics signature from 18F-fluorodeoxyglucose positron-emission tomography (PET)/CT data to determine tumor grade [42]. On the basis of the area under the curve values, the model trained on PET/CT data appeared to accurately detect tumor grade, but the overall approach is expensive to implement because of its reliance on 18F-fluorodeoxyglucose PET imaging.

In addition to the development of machine learning algorithms to analyze CT data, our group has developed an algorithm for the cytological analysis of EUS-FNA samples [43]. The goal of developing the algorithm was to create a tool that can act as a substitute for or aid with the rapid on-site evaluation (ROSE) of cytology results obtained from EUS-FNA, as ROSE is currently unavailable in many hospital settings. Our algorithm, termed Mathematical Technology for Cytopathology (MTC), does not require training data or high-powered computing resources. Instead, cytological features such as cell boundaries and nuclear status are rendered for AI analysis by converting these features into structured data. Univariate and multivariate analyses demonstrated that MTC facilitated the clinical differentiation of adenocarcinoma from benign pancreatic tissues.

Overall, machine learning has the potential to dramatically change how pancreatic cancer is diagnosed. Moreover, if the cost and turnaround times for processing CT data can be controlled, imaging-based screening could potentially be used to assess larger asymptomatic populations. However, machine learning has potential drawbacks that need to be carefully considered. Importantly, careful consideration of the data sources used to train the machine learning tools is required, and the optimal training method is currently unknown and may be disease-specific; for example, underlying genetic and/or environmental factors may limit the feasibility of using an algorithm designed to detect pancreatic cancer and trained on Western patients in Asian or African countries. Conversely, an algorithm trained on a global population may have less utility in real-world settings compared with those trained on datasets from local communities. Ongoing research is dedicated to addressing these aspects of applying machine learning to cancer diagnostics in general and will be helpful for enhancing the performance of these tools for early stage pancreatic cancer specifically.

### 3.2. Innovations in Biomarkers and Biopsy Techniques

Regarding large-scale screening programs to increase the diagnosis of early stage PDAC, innovations in imaging/radiomics may be limited by the cost of these procedures and their availability; therefore, lower-cost and higher-throughput screening methods could become important for identifying individuals who might benefit from more intensive screening. Recent advances have been made in evaluating serum biomarkers that may have utility in pancreatic cancer, and these are particularly promising as screening tools because they are low-cost and easy to use, unlike imaging methods that generally require specialized equipment and trained personnel to carry out. Most new serum screening programs rely on combining a new serum biomarker with CA19-9 and/or CEA. Novel serum biomarkers that have been recently identified largely fall into one of the following categories: cell-free DNA, cell-free RNA, cell-free proteins, circulating tumor cells, and exosomes. Markers that increase the diagnostic utility of CA19-9 include endostatin and collagen IV [44], circulating tumor-associated autoantibodies [45], methylated HOXA1 and SST [46], and pancreatic elastase-1, amylase, and lipase [47]. Moreover, new markers such as mitochondrially derived DNA and lipids in exosomes have been found in patients with PDAC [48]. The high sensitivity of detecting mutated mitochondrial DNA in exosome samples from serum suggests that this or related techniques could play an important role in future screening for pancreatic malignancies. Recent evidence indicates that tumor-derived exosomes have a wealth of unique signatures that could be used for early diagnosis [49,50]. Thus, there are high expectations for using microvesicles isolated from serum or FNA to diagnose pancreatic cancer. A recently published review by Bararia et al. provides a comprehensive description of the novel biomarkers that have the potential to be used as noninvasive means of discriminating pancreatic cancer from precancerous lesions, thereby improving the diagnosis of early stage PDAC [51].

Regarding biopsy techniques, recent advancements in EUS-FNA/FNB are also expected to improve diagnostic performance. Recent studies have begun to systematically address the optimal techniques, needle types, and additional methods and contrast agents used during EUS-FNA/FNB so that future biopsy samples are of the best quality for subsequent analysis [52,53,54]. Recently, there has been a growing preference for EUS-FNB over EUS-FNA as a means of obtaining tissue samples [55,56,57]. This shift can be attributed to reports suggesting that EUS-FNB delivers more consistent diagnostic results after improvements were made to the puncture needle [58,59]. Nonetheless, while EUS-FNB is useful in diagnosing large masses, it presents challenges in acquiring tissue fragments from smaller masses (i.e., <1 cm in diameter). In such scenarios, cytology via EUS-FNA with ROSE may offer greater utility. Mie et al. reported a study demonstrating the high diagnostic yield and safety of EUS-guided tissue acquisition for ROSE from small, solid pancreatic lesions [60]. Fitzpatrick et al. reported the diagnostic performance of cytopathology (CP) in assessing pancreatic EUS-FNB specimens, examined FNB performance based on tissue triage, and reviewed different specimen types [61]. Their findings indicated accurate diagnosis of pancreatic FNB specimens through CP, improved diagnostic yields and operating characteristics through ROSE review with CP, and enhanced overall diagnostic performance through concurrent assessment of cytological features in direct smears and architectural features in core biopsies. These results underscore the importance of CP in evaluating FNB specimens to assess sample adequacy and provide a preliminary diagnosis during the procedure. Consequently, training cytologists to perform ROSE rapidly and accurately is of utmost importance. MTC is anticipated to serve as a valuable diagnostic aid in clinical practice and as an instructional tool for cytologist training [43]. 

To date, no studies have employed machine learning or deep learning techniques to support the cytological analysis of pancreatic tissues for diagnosing adenocarcinoma in pancreatic EUS-FNA specimens. Cytological analysis is based on the morphological abnormalities of individual cells, such as nuclear atypia, while histological analysis is based on the assessment of both cellular and structural atypia and often involves samples with mixed cell types and low volumes of pathological material. In 2021, Naito et al. reported the first application of deep learning to detect adenocarcinoma in pancreatic EUS-FNB histological specimens based on training sets by expert pancreatic pathologists [62], and they noted that this approach yielded highly accurate diagnoses despite the complexity of the samples. Molecular subtyping of EUS-FNA/FNB specimens can also facilitate early diagnosis, with existing evidence suggesting the diagnostic and prognostic significance of testing samples for *KRAS* and *TP53* mutations [63,64].

## 4. Improvements to Current Surveillance Approaches

Despite decades of research on pancreatic cancer, there are still few methods available for identifying populations at higher risk of developing lesions. Conditions such as diabetes, obesity, and branch-duct intraductal papillary mucinous neoplasms (IPMNs) are known risk factors for developing pancreatic cancer; however, even with increased screening among these populations, it can be difficult to diagnose PDAC at the early stages. Therefore, there is the opportunity for important improvements to the current surveillance approaches to PDAC.

### 4.1. Insight into Indicators of Early Stage Pancreatic Cancer from Population-Based Analyses

Several recent studies have retrospectively and prospectively analyzed the features of patients with early stage pancreatic cancer with the aim of increasing the percentage of early stage diagnoses [31,65,66]. Together, these studies have identified a number of signs and risk factors that could help lead to improvements in early diagnosis.

The risk of pancreatic cancer in smokers is estimated to be 1.7 to 1.8 times higher than that in nonsmokers (Table 2). It has also been reported that the risk decreases with smoking cessation, but that it takes 20 years before the risk becomes the same as that of nonsmokers. The relative risk of pancreatic cancer due to alcohol consumption is estimated to increase 1.1- to 1.3-fold at an alcohol intake level of 24 to 50 g/day or higher. The risk of pancreatic cancer is between 1.6 and 2.0 times higher in diabetic patients. However, the risk of pancreatic cancer is 6.7-fold higher for patients with a diabetes onset of less than 1 year, 1.9-fold for those with onset between 1 and 4 years, 1.7-fold for those with onset between 5 and 9 years, and 1.4-fold for those with an onset of >10 years, indicating that the risk is higher for newly onset diabetes. Additionally, approximately 40% of patients with pancreatic cancer were reported to have developed diabetes within 5 years of their cancer diagnosis [67]. In fact, new-onset diabetes or the worsening of diabetes control is a relatively common sign of pancreatic cancer, and the possibility of pancreatic cancer should be kept in mind in these situations. The risk of pancreatic cancer is 1.34 times higher in obese individuals (body mass index ≥30 kg/m^2^ is defined as obesity). According to several meta-analyses, the increased risk of pancreatic cancer due to chronic pancreatitis is 13.3- to 16.2-fold. These reports indicate that the risk of pancreatic cancer is higher in patients with early diagnoses of chronic pancreatitis, with decreased risk in long-term cases (16-fold within 2 years, 7.9-fold within 5 years, and 3.5-fold within 9 years from the diagnosis of chronic pancreatitis). One possible explanation for this is that pancreatitis may be caused by tumor-related pancreatic duct obstruction. Although chronic pancreatitis has been shown to be a risk factor for pancreatic cancer, and long-term follow-up is considered necessary, no report has demonstrated the early detection of pancreatic cancer or improvement in prognosis via follow-up for chronic pancreatitis.

The annual incidence of pancreatic cancer is reported to be 0.95% in patients with pancreatic cysts, and the risk of developing pancreatic cancer is 22.5 times higher in these patients than in patients without cysts [68]. IPMNs, which are the most frequent type of pancreatic cysts, confer two types of cancer risk: IPMN-derived cancer, in which the IPMN directly becomes cancerous; and IPMN comorbid cancer, in which normal-type pancreatic cancer develops separately from the IPMN. In a domestic survey from Japan, half of IPMN-derived cancers were of the main pancreatic duct type, and 90% of IPMN comorbid cancers were of the branched type [69]. The average frequency of invasive carcinoma or high-grade dysplasia in main-duct-type IPMNs is 61.6% (36–100%); resection is strongly recommended in cases with a main duct diameter ≥10 mm, jaundice, and wall nodules. Conversely, the incidence of carcinoma of origin in branched IPMN has been reported to be between 0.2% and 3.0% per year and that of comorbid carcinoma to be 0% to 1.1% per year. Although IPMNs of the branched type are often followed up without high-risk findings at the time of initial diagnosis, half of all cancers of IPMN origin and 90% of comorbid cancers occur in cases of branched IPMNs, meaning the follow-up methods are more important in cases of branched IPMNs [69,70,71]. The method described in the IPMN International Practice Guidelines of 2017 [70] for the follow-up of nonresected IPMN cases is widely used, but it should be noted that this method does not consider the occurrence of comorbid cancers, and there is no recommendation for how long patients should be followed up. The 2015 American College of Gastroenterology guidelines on asymptomatic neoplastic pancreatic cysts [72] proposed that follow-up should be terminated after 5 years if there is no change in the size or characteristics of the cysts. However, a recent large-scale study from Japan showed that 68 patients (38 with IPMN-derived cancer and 30 with comorbid cancer) among 1404 total patients (9231 patient-years, median follow-up: 6.0 years) developed pancreatic malignancies, with an annual incidence rate of 0.7% and a cumulative incidence rate of 3.3% at 5 years, 6.3% at 10 years, and 15% at 15 years [31]. These results suggest that IPMN of the branched type may require longer follow-up, at least 5 years after diagnosis. In the same study, it was also reported that the cyst diameter and main pancreatic duct diameter of IPMN were associated with the incidence of carcinoma of origin but not with the incidence of comorbid cancer, suggesting that the risk of comorbid cancer should not be underestimated even in patients without a large cyst diameter or main duct diameter. Future studies should clarify the risk of developing IPMN comorbid cancer and the appropriate period of observation as well as help establish a better follow-up method. Because invasive pancreatic ductal carcinoma is a tumor originating from the pancreatic ductal epithelium, it may cause changes (indirect findings) in the pancreatic duct even at very early stages. The hazard ratio for the development of pancreatic cancer was reported to be 6.4-fold higher for even a slight dilation of the main pancreatic duct (≥2.5 mm) [73].

Another interesting finding from a population-based study was the different incidence rates of pancreatic malignancies in patients with long-standing versus new-onset diabetes [74]. Thus, while there are several conditions/ailments that indicate a higher risk for pancreatic cancer, including obesity and diabetes, a more nuanced and detailed look at these cases and their presentations could have implications for the entire PDAC patient population. For example, the genes involved in and clinical features of diseases and hereditary conditions associated with increased risk of PDAC could offer valuable insights into all cases of pancreatic cancer (Table 2).

### 4.2. Familial Pancreatic Cancer and Genetic Predispositions

Many genetic predispositions for pancreatic cancer and familial diseases that manifest with pancreatic cancer or increased risk of developing pancreatic cancer have been identified in previous decades. More recently, large-scale studies have been performed to determine the effects of different surveillance methods for these patient populations [25,26,75,76,77]. Unfortunately, these studies have also highlighted the clinical challenges of treating pancreatic cancer. Even with high-level surveillance in these defined populations, the overall conclusion is that imaging-based surveillance is insufficient, at least using current approaches and analytical methods. These studies highlight the need for new, sensitive biomarkers for PDAC so that the disease can be identified when it is treatable. Importantly, a clearer understanding of hereditary diseases and genetic predispositions for pancreatic cancer will help diagnose pancreatic cancers at earlier stages, as the underlying mechanisms of transformation are similar for sporadic and hereditary cases [30].

## 5. Future Directions

Ongoing work has suggested a number of promising new avenues for improving pancreatic cancer diagnosis, although these are largely still preliminary. Recent studies have shown that both the tumor microbiome and the intestinal microbiome may be related to pancreatic cancer development [78,79]; thus, there may be ways of using these data for diagnostics. Given the large-scale nature of microbiome data and the complexity of identifying universal disease markers, machine learning could play an instrumental role in translating these preclinical findings to clinical prediction of pancreatic cancer risk and/or development. Another intriguing area of research is preclinical modalities that could be used to diagnose pancreatic cancer, such as electrochemical biosensors based on protein and microRNA biomarkers of pancreatic cancer, which could be rapidly developed over the next decade. Finally, new findings from basic cell biology (such as microvesicle-based signaling) could ultimately be developed into new diagnostic tools for PDAC.

## 6. Conclusions

PDAC is a highly malignant tumor with a poor prognosis, which is, in large part, due to low diagnostic accuracy and poor early detection rates. While early stage PDAC can be treated with curative surgery, late-stage disease is highly metastatic and refractory to treatment. Experience with other malignancies has shown that expanding the screening population is crucial for improving the detection of early stage disease and would be likely to improve outcomes for patients with pancreatic cancer. Importantly, multidisciplinary approaches including laboratory medicine, radiology, and ultrasound will be required to increase the proportion of patients diagnosed with treatable disease. Fortunately, there have been several recent advancements in imaging and biopsy techniques that could facilitate early diagnosis. Moreover, the combination of novel biomarkers with CA19-9 and CEA could improve disease surveillance for individuals at high risk of developing PDAC. Finally, there are promising new avenues for pancreatic cancer diagnostics, such as machine learning algorithms and quantitative assessments of microvesicle-derived DNA and RNA. In addition, recent research has provided fascinating insight into the role of the microbiome in PDAC development and progression, suggesting that characteristic alterations in the oral, gut, and intratumoral microbiota could be used to diagnose PDAC and even predict treatment outcomes [80,81,82]. Similarly, the ongoing development of biosensors based on technologies such as nanoribbons, microfluidics, elliptical dichroism spectrometry, and quantum dots could lead to powerful and high-throughput means of detecting relevant biomarkers to diagnose PDAC [83,84,85,86]. In the future, these technologies and continuing innovation are expected to lead to higher rates of early stage PDAC diagnosis.

## Figures and Tables

**Figure 1 biomedicines-11-01687-f001:**
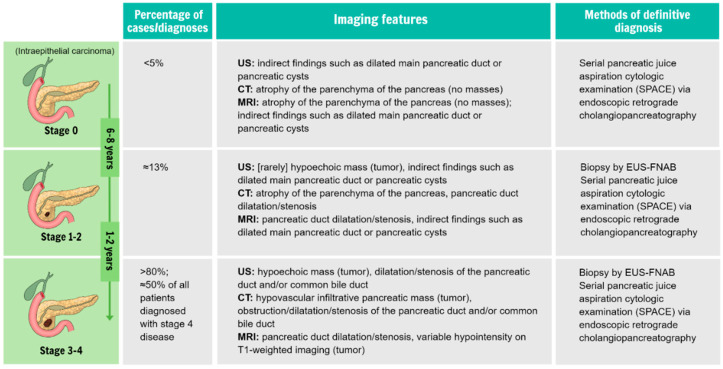
A brief summary of the stages of pancreatic ductal adenocarcinoma, with highlights for each stage regarding pathological changes that can be expected from biopsy samples and radiographical/imaging features. Abbreviations: US: ultrasound, CT: computed tomography, MRI: magnetic resonance imaging.

**Table 1 biomedicines-11-01687-t001:** Conditions and hereditary diseases associated with a high risk of developing pancreatic cancer.

	Name	Associated Risk	Gene(s) Involved	Associated Pancreatic Cancer	References
Conditions	Branch-duct intraductal papillary mucinous neoplasms	19–30% of cases progress to malignancy	*KRAS/BRAF* *PI3K/AKT* *TP53* *SHH* (hypermethylation)	Mucin-producing neoplasms	[31]
	Obesity	≈20% increased lifetime risk	*KRAS* *TP53*	PDAC	[25]
	Diabetes	≈82% increased lifetime risk	*KRAS* *GCKR*	PDAC	[23,32]
Hereditary diseases	Peutz–Jeghers syndrome	≈36% increased lifetime risk	*STK11 (LKB1)*	PDAC	[1]
	Hereditary pancreatitis	≈50% increased lifetime risk	*PRSS1*	PDAC	[1]
	Ataxia telangiectasia	≈10% increased lifetime risk	*ATM*	PDAC	[1]
	Fanconi anemia (DNA repair pathway)	At least 50-fold higher lifetime risk	*RAD51* *BRCA2* etc.	PDAC	[25]
	Lynch II syndrome	<5% increased lifetime risk	*MLHL* *MSH2/MSH6, PMS2* *EPCAM*	PDAC	[25]

Abbreviations: PDAC: pancreatic ductal adenocarcinoma.

**Table 2 biomedicines-11-01687-t002:** The relative risk of pancreatic cancer in different populations.

Factors		Risk
Hereditary disease	Peutz–Jeghers syndrome	132–140× (RR), incidence: 11–29%
	Hereditary pancreatitis	87× (SIR), incidence: 10% (50 y.o.), 50% (75 y.o.)
	Familial atypical multiple nevi melanoma syndrome	Incidence: 17% (75 y.o.)
	Hereditary breast cancer/ovarian	*BRCA1* mutation: 2.3–2.8× (RR) *BRCA2* mutation: 3.5× (RR), incidence: 4%
	Lynch syndrome	8.6× (RR), 4.5–11× (SIR), incidence: 4%
	Familial adenomatous syndrome of the colon	4.5× (RR)
Family history	Sporadic pancreatic cancer	1st degree relative with pancreatic cancer (1 person): 1.5–1.7× (RR)
	Familial pancreatic cancer	1st degree relative with pancreatic cancer (1 person): 4.5×; (2 persons): 6.4×; (3 persons): 32× (RR)
Lifestyle	Smoking	1.7–1.8× (RR)
	Alcohol use	1.1–1.3× (RR) (alcoholic consumption >24–50 g/day)
Medical history	Diabetes	1.6–2.0× (onset <1 year: 6.7×, 1–4 years: 1.9×, 5–9 years: 1.7×, >10 years: 1.4×) (RR)
	Obesity	1.3–1.4× (RR)
Pancreatic disease/imaging findings	Chronic pancreatitis	13.3–16.2× (EEs) (especially within 2 years of diagnosis)
	Pancreatic cysts	22.5× (SIR)
	IPMN	Cancer derived from IPMN: annual rate 0.2–3.0%, coexisting cancer with IPMN: annual rate 0–1.1%
	Dilation of main pancreatic duct	6.4× (main pancreatic duct ≥2.5 mm) (HR)

Abbreviations: RR: relative risk, SIR: standardized incidence ratio, y.o.: years old. HR: hazard ratio, EEs: effect estimates, IPMN: intraductal papillary mucinous neoplasm.

## Data Availability

Not applicable.

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
