# Peer review of "Advances in the Early Diagnosis of Pancreatic Ductal Adenocarcinoma and Premalignant Pancreatic Lesions"

_biomedicines, 2023, doi:10.3390/biomedicines11061687_

Round 1
Reviewer 1 Report
In this review, the authors summarize the current approaches for diagnosis and surveillance of pancreatic ductal adenocarcinoma. They also describe the leads that clinicians are testing to improve those approaches.
The review is overall well written; however, as such, the review does not sufficiently take into account novel leads for improving PDAC diagnosis and surveillance. Hence, in my opinion, before publication, there are key points that need to be improved, to reach the sufficient balancing required in a literature review.
1- There are some paragraphs that are too vague or not sufficiently back-up by literature citation; this includes (not exhaustive list : l38 to 58; 59 to 69; 170-177; 209-224; 390-394).
2- 399-402: the importance of basic research is too succinctly cited and should be mentioned earlier in the review (for example in each paragraph) as it is in contradiction with the title of the review. Mention novel fields of development such as mechanics measurements.
3- The novel leads of improvements by clinicians are not explained in details (non exhaustive: l86 “a dedicated CT protocol”: provide precisions; develop the theory behind AI applied for PDAC diagnosis; explain the difference between machine learning and deep learning that could help the reader to understand the differences between all the studies cited and their pertinence; l191 “texture feature” is taken in example; what are the other features that can be used? Why cite only this one; to which characteristics of the tissue does it correspond? (cell composition?); l232 explain why those are the most promising?; l290 what is indirect imaging? Why is it helping?; paragraph 3.2 list some new leads for biomarkers; l399 to what type of biosensor are you referring? Too vague).
4- Mention that there are other risk factors.
5- Given the current state of knowledge, stating that PDAC only comes from duct cells is too strong; several hypotheses for the origin of pancreatic ductal cancer cells are still possible and are strongly supported by experimental data / Human data (duct origin vs plasticity of other exocrine cell type that transdifferentiate into duct cells). Cite the process of acinar-to-ductal metaplasia in introduction. It is all the more important for example in the case of branched IMPN & co-morbid cancer, where plasticity of the tissue is one the most promising hypothesis that could explain this pathological process.
6- Discuss about the localisation of the tumor? Head vs Tail, there are also novel findings using spatial transcriptomics that suggest that those are different, should it change the imaging method?
7- Paragraphs that I do not understand and need to be improved (too generalized or no sufficient details to understand): l120-126 – l199-208- l272-278 – l360-362
Author Response
In this review, the authors summarize the current approaches for diagnosis and surveillance of pancreatic ductal adenocarcinoma. They also describe the leads that clinicians are testing to improve those approaches.
The review is overall well written; however, as such, the review does not sufficiently take into account novel leads for improving PDAC diagnosis and surveillance. Hence, in my opinion, before publication, there are key points that need to be improved, to reach the sufficient balancing required in a literature review.
Author response: Thank you for the positive feedback and constructive comments regarding ways to improve our manuscript.
1- There are some paragraphs that are too vague or not sufficiently back-up by literature citation; this includes (not exhaustive list : l38 to 58; 59 to 69; 170-177; 209-224; 390-394).
Author response: Thank you for your comment. The text has been revised throughout to improve the clarity and specificity, and additional citations have been added where needed.
2- 399-402: the importance of basic research is too succinctly cited and should be mentioned earlier in the review (for example in each paragraph) as it is in contradiction with the title of the review. Mention novel fields of development such as mechanics measurements.
Author response: Thank you for this comment. However, we are somewhat unclear about the intended meaning of the comment, as the cited passage refers to the need for more effective surveillance of individuals at risk of developing pancreatic cancer, which in our opinion does not contradict the title of the review. It is also somewhat unclear to us whether “mechanics measurements” refers to assessment of the mechanical stiffness of pancreatic tumor tissue, or perhaps some other diagnostic approach? This entire section has been revised to enhance its clarity, and we hope that these revisions have addressed your concerns.
3- The novel leads of improvements by clinicians are not explained in details (non exhaustive: l86 “a dedicated CT protocol”: provide precisions; develop the theory behind AI applied for PDAC diagnosis; explain the difference between machine learning and deep learning that could help the reader to understand the differences between all the studies cited and their pertinence; l191 “texture feature” is taken in example; what are the other features that can be used? Why cite only this one; to which characteristics of the tissue does it correspond? (cell composition?); l232 explain why those are the most promising?; l290 what is indirect imaging? Why is it helping?; paragraph 3.2 list some new leads for biomarkers; l399 to what type of biosensor are you referring? Too vague).
Author response: Thank you for pointing out a number of areas in which more detail was needed. We have made revisions throughout, including those that you highlighted specifically, as follows:
l86 “a dedicated CT protocol”: provide precisions;
A sentence has been added that describes the design and objective of this protocol in more detail.
develop the theory behind AI applied for PDAC diagnosis; explain the difference between machine learning and deep learning that could help the reader to understand the differences between all the studies cited and their pertinence;
The first paragraph in Section 3.1 has been revised to more clearly explain how AI is being applied to PDAC diagnosis and to describe the difference between machine learning and deep learning.
l191 “texture feature” is taken in example; what are the other features that can be used? Why cite only this one; to which characteristics of the tissue does it correspond? (cell composition?);
The second paragraph in Section 3.1 has been revised to clarify that this study looked exclusively at texture features (no other features were assessed in this study). Given the complexity of the tissue characteristics reflected by the 62 texture features explored in this study, we opted not to include a detailed description of these features in our review.
l232 explain why those are the most promising?;
The first paragraph of Section 3.2 has been revised to clarify why serum markers are promising candidates for PDAC screening.
l290 what is indirect imaging? Why is it helping?;
This paragraph was revised to provide a more suitable introduction to this section and no longer references indirect imaging; the mention of indirect findings later in the same section has been maintained.
paragraph 3.2 list some new leads for biomarkers;
Thank you for this suggestion. This paragraph has been revised to provide a more detailed description of the types of novel biomarkers that have been discovered recently and/or are currently under investigation, and to direct readers to a new review published in April that presents a comprehensive overview of novel biomarkers for PDAC (Section 3.2; Ref No. 51).
l399 to what type of biosensor are you referring? Too vague.
Thank you for this comment. The final paragraph of Section 5 has been revised to provide more detail regarding the types of biosensors that are currently under development for detecting early-stage pancreatic cancer.
4- Mention that there are other risk factors.
Author response: Thank you for this comment. The first paragraph of Section 2.2 has been revised to indicate that the list of risk factors provided in the text and in Table 1 is not exhaustive.
5- Given the current state of knowledge, stating that PDAC only comes from duct cells is too strong; several hypotheses for the origin of pancreatic ductal cancer cells are still possible and are strongly supported by experimental data / Human data (duct origin vs plasticity of other exocrine cell type that transdifferentiate into duct cells). Cite the process of acinar-to-ductal metaplasia in introduction. It is all the more important for example in the case of branched IMPN & co-morbid cancer, where plasticity of the tissue is one the most promising hypothesis that could explain this pathological process.
Author response: Thank you for this comment. The second paragraph of the Introduction has been revised to mention the potential alternative cellular origins of PDAC.
6- Discuss about the localisation of the tumor? Head vs Tail, there are also novel findings using spatial transcriptomics that suggest that those are different, should it change the imaging method?
Author response: Thank you for this suggestion. While we agree that recent molecular studies have yielded important insight into biological differences between tumors of the pancreatic head and tail, given that the focus of this review is early identification of patients with or at risk of developing pancreatic cancer, we have elected not to include a discussion of the nuances of imaging needed to properly assess a diagnosed tumor.
7- Paragraphs that I do not understand and need to be improved (too generalized or no sufficient details to understand): l120-126 – l199-208- l272-278 – l360-362
Author response: Thank you for this comment. The text throughout the manuscript has been revised to enhance its clarity and readability, with a special focus on the passages mentioned here.
Reviewer 2 Report
This is a well-written review paper dealing with overall related topics.
1. It would be better to have a more detailed review of new biomarkers. I recommend that you add a table.
2. Table 2 shows that the risk of pancreatic cancer in pancreatic cysts is 22.5x (SIR). It seems to be high. Please correct it with more references. Also, in family history, the risk of familial pancreatic cancer in 1st degree relative with pancreatic cancer (1 person) is 4.5x. Generally, familial pancreatic cancer implies more than 2 people with pancreatic cancer in 1st degree relatives. What is the meaning of 1 person?
Minor editing of English language required.
Author Response
Reviewer #2
This is a well-written review paper dealing with overall related topics.
Thank you for the positive feedback and constructive comments regarding ways to improve our manuscript.
- It would be better to have a more detailed review of new biomarkers. I recommend that you add a table.
Thank you for this suggestion. Given the wide variety and large number of recently discovered biomarkers for PDAC, we feel that this level of detail may not be appropriate for our review article. We have therefore expanded this section slightly to provide a more detailed description of the types of biomarkers that are currently under investigation, and directed readers to a new review published in April that provides a comprehensive description of the many new candidate biomarkers (Section 3.2; reference 51).
- Table 2 shows that the risk of pancreatic cancer in pancreatic cysts is 22.5x (SIR). It seems to be high. Please correct it with more references.
Thank you for this question. The value cited in this table was obtained from the sources listed below. Please note that we have not added the references to the table itself, for the sake of consistency with the rest of the table. However, they are discussed and cited in Section 4.1 of the manuscript (references 70,71,72).
- Yamaguchi, K.; et al. Pancreatic ductal adenocarcinoma derived from IPMN and ductal adenocarcinoma concomitant with IPMN. Pancreas. 2011, 40, 571–580.
Also, in family history, the risk of familial pancreatic cancer in 1st degree relative with pancreatic cancer (1 person) is 4.5x. Generally, familial pancreatic cancer implies more than 2 people with pancreatic cancer in 1st degree relatives. What is the meaning of 1 person?
Thank you for this question. For the purposes of this table, pancreatic cancer in any first-degree relative (even only a single first-degree relative) was considered to represent a family history of pancreatic cancer. We have therefore not made any revisions to this table. Please note that we have retained the information pertaining to the increase in risk associated with having more than one first-degree relative with pancreatic cancer.
- Minor editing of English language required.
Thank you for this comment. The manuscript has been thoroughly revised by a native speaker to improve the English usage.
Round 2
Reviewer 1 Report
The authors sufficiently answered to my previous questions/comments.